# SOLVING THE TRAVELING SALESMAN PROBLEM WITH POSITIONAL ENCODING

## ABSTRACT

We propose transformer-based neural solvers for the Euclidean Traveling Salesman Problem (TSP) that rely on positional encodings rather than coordinate projections. By adapting ALiBi and RoPE, modern positional encodings originally developed for large language models, to the Euclidean setting, our **Positional Encoding-based Neural Solvers (PENS)** inherit useful invariances and locality biases. To address the increased density of large instances, we introduce a simple yet effective rescaling of city coordinates that further boosts performance. Trained only on TSP-100, PENS achieves **state-of-the-art results for instances with up to 10 000 cities**, a scale that was previously dominated by methods requiring graph sparsification. These findings demonstrate that positional encodings provide effective inductive biases for neural combinatorial optimization.

## 1 INTRODUCTION

Generalizing to large-scale Euclidean Traveling Salesman Problem (TSP) instances remains a challenge for current neural combinatorial optimization (NCO) solvers. To cope with this difficulty, recent methods sparsify the input graph and restrict the decision-making to each node's nearest neighbors. While effective, this departs from the original goal of NCO: learning heuristics without hard-coded structures. Ideally, a strong solver should rely on minimal priors about the problem, giving the neural network the flexibility to learn powerful heuristics.

Most NCO solvers adopt a transformer architecture and begin by projecting raw city coordinates into the hidden dimension of the model. This mirrors how early positional encoding was done in transformers. Motivated by this observation, we explore the benefits of recent positional encoding methods, namely

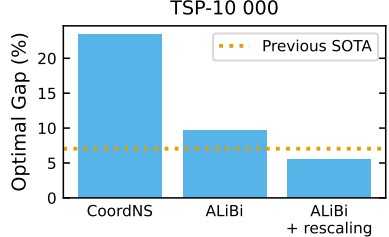

Figure 1: On TSP-10 000, PENS-A (our ALiBi-based solver) outperforms both our coordinate baseline and the previous state of the art, while requiring no input sparsification.

ALiBi (Press et al., 2022) and RoPE (Su et al., 2024), as a means for the model to better capture spatial relationships between cities. While these methods were developed for large language models initially, we demonstrate that incorporating these encodings yields consistent improvements over coordinate-based baselines, on both small- and large-scale instances. We refer to our approach as **Positional Encoding-based Neural Solvers (PENS)**.

Large-scale instances also present a challenge due to the dense spatial distribution of cities in the unit square, making them difficult for neural solvers to distinguish (Fang et al., 2024). We show that applying a simple rescaling of the city coordinates, that is, stretching the input space by an appropriate factor, substantially improves model performance. On instances with 10 000 cities, this adjustment alone divides the optimality gap by two. When combined with ALiBi positional encoding, it enables our pure transformer model to outperform the state-of-the-art sparsification-based neural solver on large-scale TSP instances.

In summary, our key contributions are as follows.

1. We introduce modern positional encoding methods, inspired by recent advances in Natural Language Processing (NLP), to represent TSP inputs within transformer-based neural solvers.

2. We demonstrate that stretching the input space through coordinate rescaling significantly improves solver performance on large-scale instances.

3. We achieve state-of-the-art results on both small and large TSP instances, without relying on sparsification or handcrafted heuristics.

The paper is organized as follows. Section 2 reviews related work, Section 3 provides background on the TSP and positional encodings, and Section 4 introduces our approach with ALiBi, RoPE, and coordinate rescaling. Section 5 reports results and ablations, and Section 6 concludes.

## 2 RELATED WORK

Neural approaches to the TSP began with Vinyals et al. (2015), who introduced Pointer Networks trained in a supervised fashion on optimal tours. Kool et al. (2019) later combined the same architecture with reinforcement learning, achieving an average $4.53\%$ gap on TSP-100 instances. Joshi et al. (2022) emphasized the importance of problem-size generalization, while Fang et al. (2024) highlighted interference from irrelevant nodes and embedding aliasing, which hinder scalability to larger TSPs.

While the TSP is defined on a **complete graph**, a common strategy is to sparsify the input to facilitate training and inference (Fu et al., 2021; Qiu et al., 2022; Sun & Yang, 2023). Approaches include local/global policies (Gao et al., 2024; Fang et al., 2024) and anchor compression (Wen et al., 2025). While sparsification helps at large scales, it is problem-specific and may degrade performance on smaller instances. For instance, Zhou et al. (2025) showed that the optimal number of neighbors depends on instance size. Motivated by these findings, we focus on neural solvers for complete TSP graphs. Prior work in this setting includes Drakulic et al. (2023) and Luo et al. (2023), who train transformers to construct tours step by step, with Drakulic et al. (2023) solving the path-TSP to enforce invariance to past decisions. We advance this line of work by introducing positional encodings as the input mechanism.

Because all cities are interconnected, **self-attention** naturally fits the TSP representation. Several works adapt attention to bias local interactions: Jin et al. (2023); Gao et al. (2024); Wang et al. (2025) add distance-dependent terms to the last attention layer, while Xiao et al. (2025a) modulate attention logits based on problem size. The latter closely resembles concurrent developments in NLP (Nakanishi, 2025).

Transformers require **positional encodings** to perceive input order. Since Vaswani et al. (2017), many alternatives have been proposed. In particular, ALiBi (Press et al., 2022) and RoPE (Su et al., 2024) bias attention logits using relative positions, and have become the standard in large language models such as BLOOM (Le Scao et al., 2022), MPT (Team, 2023), LLaMa (Touvron et al., 2023), and Gemma (Gemma Team, 2024).

Finally, the TSP is theoretically **invariant** to translation, rotation, reflection, and rescaling of the input coordinates. Architectures that embed these invariances are known to generalize more effectively. While POMO (Kwon et al., 2020) and Sym-NCO (Kim et al., 2022) enforce invariance implicitly through data augmentation and regularization, and Ouyang et al. (2024) use relative coordinates to capture translation invariance, using the distance matrix offers a representation that is intrinsically invariant. Several works have adopted this input representation (Kwon et al., 2021; Georgiev et al., 2024; Pan et al., 2025). Notably, Kwon et al. (2021) also propose modifying the transformer attention layer to embed the distance matrix, sharing similar motivation to our work. We advance this direction by designing invariant neural solvers through the specific lens of modern positional encodings.

Figure 2: Using Positional Encoding to solve Path-TSP.

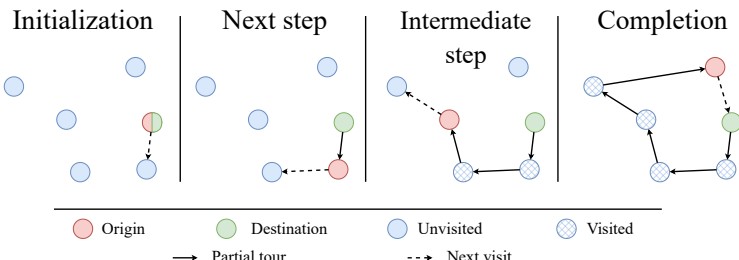

(a) **Path-TSP** Illustration on an instance with 6 cities. At each step, the solver extends the partial tour by predicting the next city, after which the origin is updated. The process continues until all cities are visited, and the tour is closed by reaching the destination.

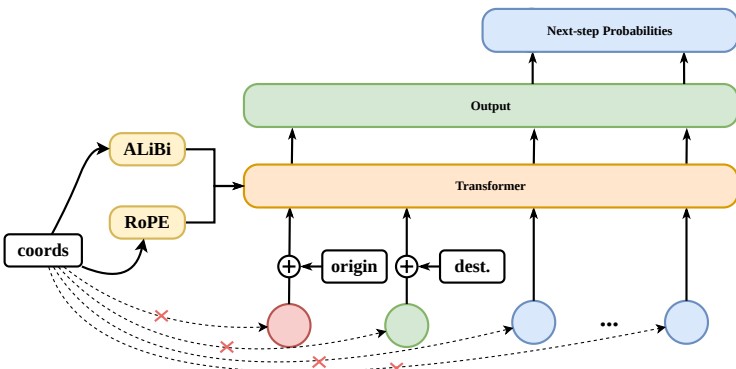

(b) **Model design** Each city is initialized with a random Gaussian embedding, while spatial information is injected through ALiBi or RoPE. Origin and destination are marked with dedicated learnable embeddings. The next city is predicted by scoring each candidate via a dot-product between the origin and the candidate embeddings.

## 3 BACKGROUND AND MOTIVATION

In this section, we first introduce the traveling salesman problem and known issues of neural solvers when generalizing to larger instances. We then present the original attention operation and the positional encodings used in our method.

### 3.1 TRAVELING SALESMAN PROBLEM IN NCO

The classical traveling salesman problem (TSP) is defined on a set of $N$ cities $\{\boldsymbol{x}_i\}_{i=1}^{N}$ in the 2D Euclidean plane. The goal is to find the shortest cycle that visits all cities exactly once. We represent the coordinates in a matrix $\boldsymbol{X} \in \mathbb{R}^{N \times 2}$. Random instances are generated by sampling city locations uniformly in the unit square.

**Path-TSP** Path-TSP (Drakulic et al., 2023) generalizes the TSP by designating an origin and a destination, $o$ and $d \in \{1, \ldots, N\}$. The objective is to find the shortest path that visits all cities once, starting at $\boldsymbol{x}_o$ and ending at $\boldsymbol{x}_d$. The original TSP is recovered when $o = d$. This formulation is particularly well suited to autoregressive neural solvers: each decision step can be expressed as a Path-TSP problem, where the model predicts the next city to visit after the current origin. After each prediction, the origin is updated to the last visited city, and previously visited cities are excluded. See Figure 2a for an illustration.

When scaling to large instances, two challenges arise during inference (Fang et al., 2024): interference from irrelevant nodes and embedding aliasing.

**Interference from irrelevant nodes**    As the number of cities increases, the self-attention mechanism (described in Section 3.2) aggregates information over many irrelevant nodes. This dilutes useful signal and makes the attention distribution less selective, preventing the model from focusing on informative neighbors.

**Embedding aliasing**    Because cities are sampled uniformly in the unit square, larger instances lead to denser configurations. In this regime, city embeddings tend to overlap, making it difficult for the model to distinguish nearby nodes. Fang et al. (2024) address this issue by introducing multiple local views, where the input space is rescaled to reduce aliasing.

## 3.2   Attention and Positional Encodings

The transformer is a neural architecture that processes a sequence of $N$ tokens, each represented by a $d$-dimensional vector, gathered in a matrix $\boldsymbol{X} \in \mathbb{R}^{N \times d}$. In natural language processing (NLP), a token typically corresponds to a subword, and the full sequence of tokens represents an input text. At each layer, tokens exchange information through the attention mechanism. From $\boldsymbol{X}$, queries $\boldsymbol{Q} = \boldsymbol{X}\boldsymbol{W}_q$, keys $\boldsymbol{K} = \boldsymbol{X}\boldsymbol{W}_k$, and values $\boldsymbol{V} = \boldsymbol{X}\boldsymbol{W}_v$ are generated, where $\boldsymbol{W}_q$, $\boldsymbol{W}_k$, and $\boldsymbol{W}_v$ are learnable projection matrices. The original self-attention operation (Vaswani et al., 2017) is defined as

$$\text{Attention}(\boldsymbol{Q}, \boldsymbol{K}, \boldsymbol{V}) = \text{softmax}\left(\frac{\boldsymbol{Q}\boldsymbol{K}^\top}{\sqrt{d}}\right)\boldsymbol{V}.$$

The matrix $\boldsymbol{Q}\boldsymbol{K}^\top$ contains the attention logits that determine how information is shared between tokens.

**ALiBi**    The transformer by default treats the tokens $\boldsymbol{X}$ as an unordered set. However, when tokens correspond to elements of a sequence, their order must be incorporated for meaningful processing. ALiBi (Press et al., 2022) is a positional encoding method that integrates token positions directly into the attention computation by adding a distance-dependent bias to the attention logits:

$$\text{Attention}(\boldsymbol{Q}, \boldsymbol{K}, \boldsymbol{V}) = \text{softmax}\left(\frac{\boldsymbol{Q}\boldsymbol{K}^\top}{\sqrt{d}} - m\boldsymbol{D}\right)\boldsymbol{V},$$

where $\boldsymbol{D}$ is the pairwise distance between token positions in the input sequence and $m > 0$ is a slope parameter. This formulation not only provides positional information, but also biases the model toward attending more strongly to nearby tokens, as the attention scores decay linearly with distance.

**RoPE**    RoPE (Su et al., 2024) is another positional encoding method based on relative positions. Instead of adding a bias to the attention logits, RoPE applies a rotation to queries and keys that depends on their position in the sequence. For a vector $\boldsymbol{x} \in \mathbb{R}^d$ at position $p$, RoPE applies a block-diagonal rotation matrix $\boldsymbol{R}^d(\Theta, p)$ composed of two-dimensional rotations applied to the $d/2$ pairs, with frequencies $\Theta = (\theta_i)_{i=1}^{d/2}$:

$$\text{RoPE}(\boldsymbol{x}, p) = \boldsymbol{R}^d(\Theta, p)\,\boldsymbol{x}.$$

Under this encoding, the attention logit between a query at position $p$ and a key at position $n$ becomes

$$\begin{aligned}
\boldsymbol{q}_p^\top \boldsymbol{k}_n &= \left(\boldsymbol{R}^d(\Theta, p)\,\boldsymbol{W}_q\boldsymbol{x}_p\right)^\top \left(\boldsymbol{R}^d(\Theta, n)\,\boldsymbol{W}_k\boldsymbol{x}_n\right) \\
&= \boldsymbol{x}_p^\top \boldsymbol{W}_q^\top\,\boldsymbol{R}^d(\Theta, n-p)\,\boldsymbol{W}_k\boldsymbol{x}_n.
\end{aligned}$$

Thus, the logits depend only on the relative offset $n - p$, making RoPE a relative positional encoding. The rotation frequencies $\Theta$ are fixed in advance and are not learned. Intuitively, positions are converted into rotation phases, so relative distances between tokens correspond to relative phase shifts in their embeddings.

**Axial-RoPE** Axial-RoPE (Heo et al., 2024) extends RoPE to two-dimensional inputs such as images, where each token has coordinates $(x, y)$. Queries and keys are split into two halves: the first half is rotated according to the $x$-coordinate and the second half according to the $y$-coordinate. The resulting vectors are then concatenated back together. This design allows RoPE to encode relative positions along both axes independently.

## 4 METHOD

We now explain how we use ALiBi (Press et al., 2022) and RoPE (Su et al., 2024; Heo et al., 2024) inside the transformer to solve the TSP. We call our neural solvers **Positional Encoding-based Neural Solvers (PENS)**.

### 4.1 INPUT PERCEPTION WITH ALiBi AND RoPE

Most NCO solvers, such as BQ-NCO (Drakulic et al., 2023), begin by projecting the $N$ city coordinates $\boldsymbol{X} \in \mathbb{R}^{N \times 2}$ into the hidden dimension $d$ of the transformer using a learnable linear transformation $\boldsymbol{W} \in \mathbb{R}^{2 \times d}$. The initial node embeddings are thus given by $\boldsymbol{XW}$.

In contrast, we do not provide raw coordinates directly to the model. Each city is first assigned a random Gaussian embedding $\mathbf{x} \sim \mathcal{N}(\mathbf{0}, \boldsymbol{I}_d)$, and its spatial information is injected exclusively through positional encodings. Specifically, we adapt ALiBi (Press et al., 2022) and RoPE (Su et al., 2024; Heo et al., 2024) to operate on city distances and coordinates, respectively. This design choice removes the reliance on coordinate projection and allows us to exploit the inductive biases of these positional encoding schemes.

This approach offers several advantages:

- **ALiBi** biases attention according to pairwise city distances, promoting invariance to translations, rotations, and symmetries while improving generalization to larger TSP instances.
- **RoPE** encodes both the magnitude and orientation of relative displacements, ensuring translation invariance and providing richer positional features than coordinate projection alone.

**ALiBi** In the standard transformer, ALiBi introduces a linear bias proportional to the relative index distance between tokens. We adapt this idea by replacing index distance with the Euclidean distance between cities. Concretely, we construct the distance matrix $\boldsymbol{D} \in \mathbb{R}^{N \times N}$ with entries $d_{ij} = \|\boldsymbol{x}_i - \boldsymbol{x}_j\|_2$. The ALiBi bias is then applied so that attention between two cities decreases as their distance grows. This mechanism softly encourages information to propagate locally.

Each attention head uses a different slope parameter $m_h$, defined as

$$m_h = \frac{10}{\sqrt{2}^h}, \quad \forall h \in \{0, \ldots, n_{\text{heads}} - 1\}.$$

Larger slopes correspond to heads focusing on short-range interactions, while smaller slopes allow long-range information flow. Because the bias depends only on pairwise distances, which are invariant to translations, rotations, and reflections, the resulting solver is natively invariant to these transformations.

**RoPE** To adapt RoPE to two-dimensional city coordinates, we use the axial formulation (Hao et al., 2024): the first half of each query and key vectors are rotated by the $x$-coordinate, and the second half by the $y$-coordinate. The rotation frequencies follow Heo et al. (2024) and are defined for the unit square as

$$\theta_i = 14 \cdot 100^{-i/(d/4)}, \quad \forall i \in \{0, \ldots, d/4 - 1\}.$$

This encoding is strictly translation-invariant, since shifting all coordinates by the same vector does not affect the angular relations between cities. Unlike ALiBi, which only modulates attention strength based on distances, RoPE preserves both the magnitude and orientation of relative

displacements between cities. As a result, it provides a more expressive representation of spatial relations, though without invariance to rotations or reflections. For simplicity, we refer to this axial formulation as RoPE throughout the paper.

## 4.2 STRETCHING THE INPUT SPACE

As mentioned in Section 3.1, embedding aliasing (Fang et al., 2024) occurs when city coordinates become densely distributed, making them difficult for a neural solver to distinguish. To mitigate this effect, we apply a uniform multiplicative scaling factor to the coordinates before passing them to the model. This rescaling reduces the chance that distinct cities are mapped to nearly identical embeddings, which can happen when cities cluster too closely. By spreading them apart, the model's attention operates in a regime where small coordinate differences are more distinguishable, improving the solver's ability to learn meaningful spatial relations.

We evaluate the impact of the scaling factor by solving random TSP instances across a range of scaling values. The results, summarized in Figure 3, show that solution quality improves as the scaling factor increases, up to a point of diminishing returns. To approximate the best scaling factor, we fit a quadratic curve to the experimental results and compute its minimum. This procedure is applied independently for each trained neural solver. Importantly, the scaling factor is tuned exclusively on held-out random instances, never on the benchmark instances used for final evaluation.

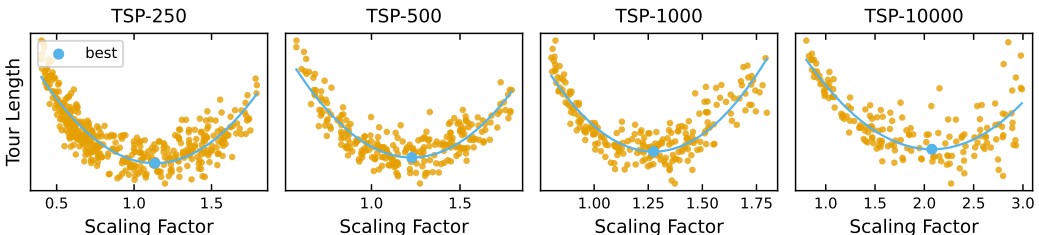

Figure 3: Performance of PENS-A, our ALiBi-based solver, when rescaling city coordinates on random TSP instances of varying sizes. Each scatter point reports the average tour length (measured on the original coordinate scale) for a given scaling factor. A quadratic curve is fitted to the results for each TSP size to estimate the optimal scaling factor. The estimated optimal scaling factor increases with the size of the instances.

## 4.3 ARCHITECTURE

Our solver follows a standard transformer backbone (Zhang & Sennrich, 2019; Xiong et al., 2020), enhanced with Scalable-Softmax (SSMax) (Nakanishi, 2025; Xiao et al., 2025a), which has been shown to improve generalization across problem sizes (Xiao et al., 2025a). Within the self-attention layers, spatial information is incorporated using either ALiBi or Axial-RoPE, as described in Section 4.1.

Each city is initialized with a random Gaussian embedding $\mathbf{x} \sim \mathcal{N}(\mathbf{0}, \boldsymbol{I}_d)$. In addition, we introduce two learnable vectors $\boldsymbol{e}_o, \boldsymbol{e}_d \in \mathbb{R}^d$ that represent the origin and destination. These vectors are added to the initial embeddings of the corresponding cities, providing explicit markers for the start and end of the tour.

At decoding time, the next city is predicted by scoring the compatibility between the hidden representation of the current origin and each remaining candidate. Let $\boldsymbol{x}_o^{(L)} \in \mathbb{R}^d$ denote the final hidden state of the current origin after $L$ layers, and let $\boldsymbol{X}_c^{(L)} \in \mathbb{R}^{N_c \times d}$ be the states of the $N_c$ remaining cities. Output logits are computed as

$$(\boldsymbol{x}_o^{(L)}\boldsymbol{W}_q)(\boldsymbol{X}_c^{(L)}\boldsymbol{W}_k)^\top = \boldsymbol{q}_o\boldsymbol{K}_c^\top \in \mathbb{R}^{N_c},$$

where $\boldsymbol{W}_q, \boldsymbol{W}_k \in \mathbb{R}^{d \times d}$ are learnable matrices, $\boldsymbol{q}_o = \boldsymbol{x}_o^{(L)}\boldsymbol{W}_q$, and $\boldsymbol{K}_c = \boldsymbol{X}_c^{(L)}\boldsymbol{W}_k$. This final step does not use positional encodings or SSMax, keeping the output layer lightweight and focused on city-to-city compatibility.

An overview of the model is shown in Figure 2b, and more details are provided in Appendix A.3.

## 5 RESULTS

We now present our main findings. We first describe the training setup, evaluation metrics, and the state-of-the-art baselines used for comparison. We then report results on both synthetic and real benchmarks, and conduct ablation studies to assess the impact of individual design choices.

### 5.1 EXPERIMENTAL SETUP

**Models** We train three model variants that differ in their positional encoding strategy: one using ALiBi, one using RoPE, and one combining both, which we denote as **PENS-A**, **PENS-R**, and **PENS-AR**, respectively. All models follow the hyperparameters of BQ-NCO (Drakulic et al., 2023): they consist of 9 transformer layers with a hidden dimension of 192, a feedforward dimension of 512, and 12 attention heads. Each model has a total of 2.8M learnable parameters. For comparison, we also train an additional baseline, **CoordNS** (Coordinates-based Neural Solver), that directly projects the raw city coordinates without any additional positional encoding mechanism.

**Training setup** All models are trained on random uniform TSP-100 instances. For each instance in a batch, we randomly select an origin-destination pair from its optimal tour, which defines a path-TSP problem. Solvers are trained to predict the next city to visit immediately after the origin. Training labels are obtained from optimal solutions computed with Concorde (Applegate et al., 2006).

We use the AdamW optimizer for 1M training steps with a batch size of 1024, using a cosine annealing learning rate schedule that decays from $10^{-4}$ to $10^{-5}$. Each model requires three days of training on a single NVIDIA RTX5090 GPU.

**Evaluation** We evaluate all methods on both randomly generated TSPs and TSPLIB (Reinelt, 1995), with problems ranging from 100 up to 10 000 cities. Optimal solutions are obtained with Concorde (Applegate et al., 2006), except for TSP-10 000, where we limit Concorde to six hours per instance.

We measure performance using the (near-)optimality gap, defined as:

$$\text{gap} = \frac{c_{\text{model}} - c_{\text{conc}}}{c_{\text{conc}}},$$

where $c_{\text{model}}$ and $c_{\text{conc}}$ are the solution costs produced by the model and Concorde, respectively. Solutions are generated autoregressively. Because the decoding strategy strongly influences the final quality (François et al., 2019; Xia et al., 2024), we adopt greedy decoding: we select the most probable city at each step, without any additional search process. To reduce variance, each instance is solved five times and we report the average gap. We also report the total runtime across all instances, using a batch size of 1.

**Baselines** We compare our models against state-of-the-art neural TSP solvers covering different problem scales. We include LEHD (Luo et al., 2023) and BQ-NCO (Drakulic et al., 2023), which operate on the complete graph and represent the state of the art for instances with up to 1000 cities. For larger instances, we evaluate INViT (Fang et al., 2024) and DGL (Xiao et al., 2025b), which sparsify the input graph and are the current state of the art on problems with 10 000 cities. For INViT, we consider both reported variants, INViT-2V and INViT-3V, which use two and three local views during inference, respectively. In addition, we evaluate BQ-NCO enhanced with the Entropy-Scaling Factor (ESF) (Xiao et al., 2025a), which has been shown to improve its performance on large-scale problems.

All baselines solvers are autoregressive, like ours. For fairness, we evaluate them under greedy decoding, ensuring that comparisons isolate model architecture rather than decoding strategy.

| Model | Uniform TSP | | | | |
|---|---|---|---|---|---|
| | 100 | 250 | 500 | 1000 | 10 000 |
| CoordNS | 0.37 | 0.67 | 1.05 | **1.51** | 23.47 |
| PENS-A | 0.34 | 0.75 | 1.22 | 1.86 | **9.74** |
| PENS-R | **0.26** | **0.55** | **0.95** | 1.75 | 16.73 |

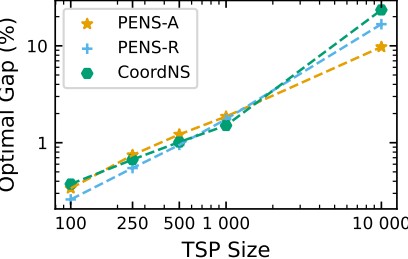

Figure 4: We compare PENS-A (using ALiBi), PENS-R (using RoPE) and CoordNS that directly uses the cities coordinates. PENS-R is the best at small-scale while PENS-A clearly leads at large-scale instances. We conclude that RoPE provides rich positional features while ALiBi prevents large-scale collapse.

## 5.2 Results

**PE effectiveness** We first compare PENS-A and PENS-R against CoordNS, our baseline projecting the raw city coordinates. Our results in Figure 4 indicate that RoPE is most effective on small-scale problems, while ALiBi dramatically improves large-scale generalization. PENS-A leverages the invariances and local bias introduced by ALiBi, yielding robustness when the self-attention mechanism involves thousands of cities. Conversely, the fact that PENS-R leads on small-scale suggests that RoPE offers rich features that are easier for the solver to exploit, but lacks a mechanism to maintain performance as the problem size grows. Thus, RoPE contributes richer positional features, whereas ALiBi offers a more robust mechanism for large-scale generalization.

| Model | Uniform TSP | | | | | | | | | |
|---|---|---|---|---|---|---|---|---|---|---|
| | 100 | | 250 | | 500 | | 1000 | | 10 000 | |
| | gap | factor | gap | factor | gap | factor | gap | factor | gap | factor |
| PENS-A | 0.34 | - | 0.75 | - | 1.22 | - | 1.86 | - | 9.74 | - |
| | 0.36 | 1.01 | 0.71 | 1.13 | 1.14 | 1.23 | 1.68 | 1.27 | 5.46 | 2.08 |
| PENS-R | 0.26 | - | 0.55 | - | 0.95 | - | 1.75 | - | 16.73 | - |
| | 0.27 | 1.00 | 0.53 | 1.03 | 0.88 | 1.08 | 1.52 | 1.15 | 11.86 | 1.24 |

Table 1: Optimality gap (in %) for multiple TSP sizes, comparing the original scale (factor 1.00) against the best estimated scaling factor. Gaps are computed with respect to the original coordinate scale. Rescaling has little effect on small instances, but substantially improves performance on larger ones.

**Rescaling the input** As mentioned in Section 3.1, embedding aliasing (Fang et al., 2024) occurs when city coordinates become too densely distributed, making them difficult for the model to distinguish. To mitigate this effect, we apply a uniform scaling factor to the coordinates before providing them to the neural solver. This simple adjustment yields a marked improvement on large instances: for example, PENS-A on TSP-10 000 reduces its optimality gap from $9.47\%$ to $5.46\%$, a nearly twofold gain that establishes it as the state-of-the-art solver at this scale (see Table 2). At smaller sizes (fewer than 1000 cities), the impact of rescaling is marginal. The complete results are provided in Table 3 in the Appendix.

**Main results** Our evaluations on uniform TSPs and TSPLIB are reported in Table 2. From a quality standpoint, PENS-R achieves state-of-the-art results on instances with less than 1 000 cities, while PENS-A dominates on instances with 10 000 cities. Notably, PENS-A outperforms INViT-3V (Fang et al., 2024), the strongest sparsity-based solver previously reported on TSP-10 000. In terms of efficiency, RoPE only introduces an additional $O(N)$ overhead, whereas ALiBi requires an additional $O(N^2)$ term. We note, however, that RoPE currently benefits from optimized flash-attention implementations (Dao et al., 2022; Dao, 2024), similar optimizations could also be applied

to ALiBi. Finally, while PENS-A and PENS-R each specialize at different scales, their combination, PENS-AR, yields consistently strong performance across all scales, as shown in Table 2.

| Model | Uniform TSP | | | | | | | | | | TSPLIB | | | | | |
| | 100 | | 250 | | 500 | | 1000 | | 10 000 | | 1∼100 | | 101∼1000 | | 1001 ∼ 10 000 | |
| | gap | time | gap | time | gap | time | gap | time | gap | time | gap | time | gap | time | gap | time |
| BQ-NCO | 0.31 | 0.6m | 0.67 | 1.5m | 1.17 | 3.8m | 2.19 | 13m | 19.94 | 245m | 0.48 | 0.1m | 2.80 | 0.6m | 11.31 | 27m |
| BQ-NCO + ESF | 0.34 | 0.6m | 0.67 | 1.6m | 1.04 | 4.0m | 1.71 | 14m | 15.12 | 322m | 0.50 | 0.1m | **2.76** | 0.6m | 8.49 | 29m |
| LEHD | 0.49 | 0.4m | 0.95 | 1.0m | 1.63 | 2.0m | 3.06 | 4m | 28.85 | 53m | 0.61 | 0.1m | 3.14 | 0.4m | 12.34 | 5m |
| INViT-2V | 4.92 | 0.9m | 5.98 | 2.3m | 6.45 | 4.8m | 6.69 | 10m | 7.08 | 20m | 6.04 | 0.1m | 8.63 | 0.8m | 10.88 | 6m |
| INViT-3V | 4.85 | 1.0m | 5.92 | 2.8m | 6.30 | 5.9m | 6.76 | 13m | 7.05 | 25m | 4.91 | 0.1m | 9.07 | 0.9m | 12.33 | 7m |
| DGL | 2.19 | 0.5m | 3.37 | 1.2m | 4.72 | 2.3m | 5.74 | 5m | 7.81 | 4m | 2.60 | 0.1m | 7.33 | 0.4m | 9.44 | 2m |
| PENS-A | 0.34 | 1.4m | 0.75 | 3.7m | 1.22 | 7.6m | 1.68 | 15m | **5.46** | 69m | 0.52 | 0.1m | 3.32 | 1.3m | 6.64 | 8m |
| PENS-R | 0.26 | 1.8m | 0.55 | 4.6m | 0.95 | 9.3m | **1.52** | 19m | 11.86 | 18m | 0.39 | 0.1m | 2.89 | 1.6m | 8.14 | 8m |
| PENS-AR | **0.20** | 1.9m | **0.46** | 4.9m | **0.94** | 9.8m | 1.56 | 20m | 7.13 | 81m | **0.34** | 0.1m | 3.14 | 1.6m | **5.51** | 11m |

Table 2: Comparison with state-of-the-art neural solvers. All models are evaluated on the same set of instances using greedy decoding. For instances with more than 1000 cities, our models use their best estimated scaling factors. We show both the optimal gap (in %) and the total time required to solve the instances. BQ-NCO (Drakulic et al., 2023), BQ-NCO + ESF (Xiao et al., 2025a) and LEHD (Luo et al., 2023) are solvers that do not sparsify the input graph while INViT-2V, INViT-3V (Fang et al., 2024) and DGL (Xiao et al., 2025b) do sparsify. Sparsification reduces performance on small instances but improves scalability to large ones. PENS-A, based on ALiBi, achieves state-of-the-art results on TSP-10 000 without sparsification. Combining ALiBi and RoPE, PENS-AR attains state-of-the-art performance across nearly all benchmarks.

**Gaussian embeddings initialization**  We assess the impact of initializing city embeddings with random Gaussian vectors against zero vectors. To this end, we train small versions of PENS-A and PENS-R on TSP-20 and evaluate them on TSP-20, TSP-50, and TSP-100.

Figure 5: Performance comparison between zero initialization and Gaussian initialization.

| Model | Uniform TSP | | |
| | 20 | 50 | 100 |
| PENS-A | | | |
| Zero-init | **0.096** | **1.629** | 4.996 |
| Gaussian-init | 0.178 | 1.649 | **4.282** |
| PENS-R | | | |
| Zero-init | 0.070 | 1.600 | 9.896 |
| Gaussian-init | **0.068** | **1.163** | **3.459** |

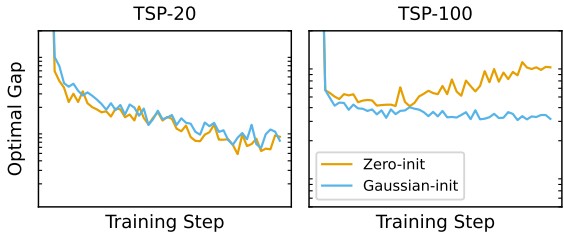

(a) Final optimality gaps (in %) on TSP-20/50/100. Gaussian embeddings improves large-scale performance.

(b) Training curves for PENS-R on TSP-20, evaluated on both TSP-20 and TSP-100. The zero-initialized variant quickly overfits to the training scale (TSP-20).

The results, summarized in Table 5a, indicate that Gaussian initialization does not affect performance on TSP-20, the training problem size. However, it significantly improves generalization to larger instances, particularly for PENS-R. Figure 5b further illustrates this effect: the zero-initialized variant quickly overfits to TSP-20, whereas Gaussian embeddings maintain progress across problem sizes.

We hypothesize that the stochasticity introduced by Gaussian embeddings promote the learning of simpler heuristics that transfer more effectively to larger instances.

## 6    CONCLUSION AND FUTURE WORK

We have proposed the use of modern positional encodings to enhance the capacity of neural TSP solvers and demonstrated the importance of rescaling city coordinates to improve distinguishability on large-scale instances. Combined, our Positional Encoding-based Neural Solvers (PENS) achieve state-of-the-art performance across a wide range of problem sizes. Importantly, our solvers operate

directly on the raw representation of the problem, avoiding the need for input sparsification, which risks oversimplifying the problem.

While these solvers are effective, they currently require a full forward pass at each decoding step, which can limit efficiency. Future work includes exploring ways to cache or reuse intermediate computations, similar to key-value caching in language models (Kwon et al., 2023), to accelerate decoding. It would also be interesting to extend ALiBi-based solvers to problems defined solely by a cost matrix, such as the asymmetric TSP, and to learn ALiBi slopes and RoPE angles in an end-to-end manner to further improve performance.

**Reproducibility statement**  The code, data and models are fully released at the provided URL (temporarily as supplementary materials during the review process). Implementation details are described in Section 5.1 and in Appendix A.3. To ensure reproducibility, the code is distributed with pinned dependencies, and the README includes the commands to generate data, train models and run evaluations. We also provide code to reproduce the baseline results from previous works.

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

# A APPENDIX

## A.1 ATTENTION VISUALIZATION

We explore how models act under different scaling factors. To this end, we fix a single TSP-10 000 instance and visualize the average attention logits of the origin city with respect to all other cities (Figure 6).

For small scaling factors (below 1.5), all models attend mainly to cities close to the origin and to the destination. At larger scaling factors, however, only PENS-A (the ALiBi-based solver) maintains consistent attention, continuing to focus on both local neighborhoods and the destination. In contrast, the baseline fails to attend to the destination, while PENS-R exhibits periodic attention to specific regions of the square, likely caused by rotations of queries and keys that constructively interfere. Similar periodic patterns have been reported in the NLP domain (Sun et al., 2022; Peng et al., 2024), where enhancements were proposed to improve context-length generalization.

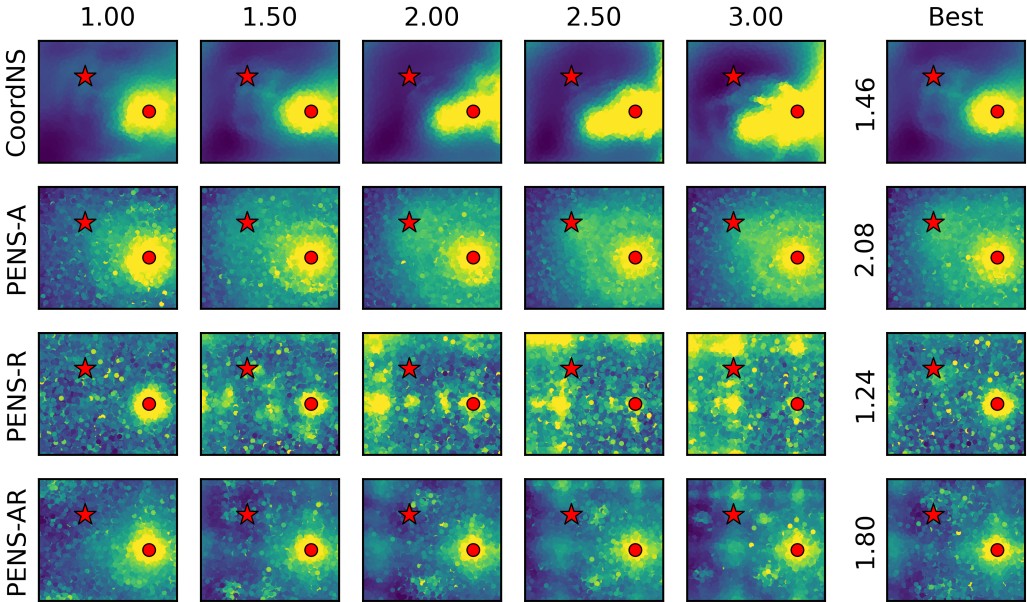

Figure 6: Attention patterns on a path-TSP-10 000 instance under different scaling factors. Each panel shows the average attention logits from the origin city (red circle) to all others. The destination is marked with a red star. The rightmost column uses the best estimated scaling factor of each model. ALiBi-based solvers remain consistent across scales, concentrating on local neighborhoods and the destination, while other models deteriorate or show periodic artifacts.

## A.2 EVALUATION DETAILS

We report all results in Table 3. Additional details about baselines are provided in Table 4. As described in Section 5.1, we evaluate all baselines using greedy decoding in order to isolate the impact of the architecture. For reference, the decoding strategies originally used in the respective papers are also listed in Table 4. All baselines are evaluated using the official code released by the authors.

## A.3 IMPLEMENTATION DETAILS

**Transformer** We use a prenorm Transformer (Xiong et al., 2020) with RMSNorm (Zhang & Sennrich, 2019), which is the most common transformer variant in current practice.

| Model | Uniform TSP | | | | | | | | | | | | TSPLIB | | | | | |
|---|---|---|---|---|---|---|---|---|---|---|---|---|---|---|---|---|---|---|
| | 100 128 inst. | | 250 128 inst. | | 500 128 inst. | | 1000 128 inst. | | 10 000 10 inst. | | | | 1~100 12 inst. | | 101~1000 36 inst. | | 1001~10 000 24 inst. | |
| | gap | time | gap | time | gap | time | gap | time | gap | time | | | gap | time | gap | time | gap | time |
| BQ-NCO | 0.31 | 0.6m | 0.67 | 1.5m | 1.17 | 3.8m | 2.19 | 13m | 19.94 | 245m | | | 0.48 | 0.1m | 2.80 | 0.6m | 11.31 | 27m |
| BQ-NCO + ESF | 0.34 | 0.6m | 0.67 | 1.6m | 1.04 | 4.0m | 1.71 | 14m | 15.12 | 322m | | | 0.50 | 0.1m | **2.76** | 0.6m | 8.49 | 29m |
| LEHD | 0.49 | 0.4m | 0.95 | 1.0m | 1.63 | 2.0m | 3.06 | 4m | 28.85 | 53m | | | 0.61 | 0.1m | 3.14 | 0.4m | 12.34 | 5m |
| INViT-2V | 4.92 | 0.9m | 5.98 | 2.3m | 6.45 | 4.8m | 6.69 | 10m | 7.08 | 20m | | | 6.04 | 0.1m | 8.63 | 0.8m | 10.88 | 6m |
| INViT-3V | 4.85 | 1.0m | 5.92 | 2.8m | 6.30 | 5.9m | 6.76 | 13m | 7.05 | 25m | | | 4.91 | 0.1m | 9.07 | 0.9m | 12.33 | 7m |
| DGL | 2.19 | 0.5m | 3.37 | 1.2m | 4.72 | 2.3m | 5.74 | 5m | 7.81 | 4m | | | 2.60 | 0.1m | 7.33 | 0.4m | 9.44 | 2m |
| PENS-A | 0.34 | 1.4m | 0.75 | 3.7m | 1.22 | 7.6m | 1.86 | 15m | 9.74 | 69m | | | 0.52 | 0.1m | 3.32 | 1.3m | 8.82 | 8m |
| PENS-A *(scaled)* | 0.36 | 1.4m | 0.71 | 3.7m | 1.14 | 7.6m | 1.68 | 15m | **5.46** | 69m | | | 0.48 | 0.1m | 3.53 | 1.3m | 6.64 | 8m |
| PENS-R | 0.26 | 1.8m | 0.55 | 4.6m | 0.95 | 9.3m | 1.75 | 19m | 16.73 | 18m | | | 0.39 | 0.1m | 2.89 | 1.6m | 8.73 | 8m |
| PENS-R *(scaled)* | 0.27 | 1.8m | 0.53 | 4.6m | **0.88** | 9.3m | **1.52** | 19m | 11.86 | 18m | | | 0.43 | 0.1m | 3.04 | 1.6m | 8.14 | 8m |
| PENS-AR | **0.20** | 1.9m | **0.46** | 4.9m | 0.94 | 9.8m | 1.75 | 20m | 9.51 | 81m | | | **0.34** | 0.1m | 3.14 | 1.6m | 6.39 | 11m |
| PENS-AR *(scaled)* | 0.22 | 1.9m | 0.50 | 4.9m | 0.89 | 9.8m | 1.56 | 20m | 7.13 | 81m | | | 0.38 | 0.1m | 3.13 | 1.6m | **5.51** | 11m |

Table 3: Results of our models and the baseline solvers. For each PENS variant, we report performance with and without input rescaling. For TSPLIB instances, the rescaling factor is estimated by linear interpolation from uniform TSP results based on instance size. Optimal gaps are shown in %.

| Method | Graph type | # Params | Venue | Decoding |
|---|---|---|---|---|
| LEHD (Luo et al., 2023) | Complete | 1.4M | NeurIPS 2023 | RRC |
| BQ-NCO (Drakulic et al., 2023) | Complete | 3.1M | NeurIPS 2023 | BS |
| BQ-NCO+ESF (Xiao et al., 2025a) | Complete | 3.1M | Neural Networks 2025 | BS |
| INViT-2V/3V (Fang et al., 2024) | Sparsified | 1.7M/2.6M | ICML 2024 | BS + POMO |
| DGL (Xiao et al., 2025b) | Sparsified | 0.8M | IJCAI 2025 | BS + POMO |
| PENS *(ours)* | Complete | 2.7M | - | - |

Table 4: Overview of the evaluated baselines. RRC denotes *Random Re-Construct*, which randomly regenerates partial solutions. BS stands for *Beam Search*. POMO refers to Kwon et al. (2020), which augments each instance with multiple starting points, solves them independently, and returns the best solution.

**Scalable Softmax**   Scalable Softmax (SSMax) (Nakanishi, 2025) was proposed for LLMs to improve long-context capabilities. A related approach, the *Entropy-based Scaling Factor* (ESF), was later introduced for neural TSP solvers (Xiao et al., 2025a). The idea is to attenuate irrelevant tokens in long sequences by increasing the signal-to-noise ratio before the weighted sum. This is achieved by multiplying the attention logits by a scaling factor $s$:

$$\text{Attention}(\boldsymbol{Q}, \boldsymbol{K}, \boldsymbol{V}) = \boldsymbol{V} \, \text{softmax}\left( s \, \frac{\boldsymbol{Q}\boldsymbol{K}^\top}{\sqrt{d}} \right),$$

where $s \propto \log(N)$ and $N$ is the sequence length. Nakanishi (2025) use a learned scaling factor, while Xiao et al. (2025a) set it according to the training instance size. In this work, we simply use

$$s = \log(N + 1).$$

**ALiBi slopes**   We initially experimented with trainable slopes but found them difficult to optimize: removing weight decay and adding a regularization loss were required to prevent convergence to zero. Empirically, the learned slopes varied between 0 and 10, which motivated us to fix them as

$$m_h = \frac{10}{\sqrt{2}^h},$$

inspired by the original ALiBi method (Press et al., 2022).

**Flex-attention**   Our self-attention layer deviates slightly from standard RoPE and ALiBi implementations, preventing the use of existing flash-attention kernels (Dao et al., 2022; Dao, 2024). We instead rely on PyTorch's flex-attention[1], which allowed us to implement these layers efficiently. Flex-attention is still in beta and required workarounds not covered in the documentation. We invite the reader to consult our released code for details. To manage memory on large instances, we also employed chunked attention (Kwon et al., 2023).

---

[1]See https://pytorch.org/blog/flexattention/.

