# OpenReview forum: "Solving the Traveling Salesman Problem with Positional Encoding"
_ICLR.cc/2026/Conference — Submitted to ICLR 2026_

### Official Review · Reviewer_TNia · 2025-10-18

**Soundness:** 2
**Presentation:** 2
**Contribution:** 2
**Rating:** 4
**Confidence:** 4

**Summary:**

This manuscript proposes Positional Encoding-based Neural Solvers (PENS) for addressing cross-scale instances of the TSP.

**Strengths:**

This manuscript is easy to follow.

**Weaknesses:**

**W1 Limited innovation:** The proposed PENS reads as a combination of several existing ideas rather than a clearly novel contribution. Prior studies [1,2] have already incorporated distance information between nodes into attention computations. In addition, an order embedding strategy has been adopted by the iteration-based NCO solver [3]. Finally, the scaling strategy has also been explored [4].

**W2 Heavy dependence on tuning the scaling factor:**  The scaling factor appears to require extensive preliminary tuning for instances of each scale to obtain good performance, which undermines the method’s practicality in real-world settings.

**W3 Insufficient experiments:**  Experiments are limited to TSP instances only. To demonstrate cross-task generalization, the authors should evaluate PENS on other routing problems, such as CVRP.


[1] Distance-aware attention reshaping for enhancing generalization of neural solvers. TNNLS, 2025.

[2] Instance-Conditioned Adaptation for Large-scale Generalization of Neural Routing Solver. arxiv, 2024.

[3] Learning to Iteratively Solve Routing Problems with Dual-Aspect Collaborative Transformer. NeurIPS, 2021.

[4] Improving generalization of neural vehicle routing problem solvers through the lens of model architecture. Neural Networks, 2025.

**Questions:**

PENS selects random origin and destination nodes. Is there a specific selection pattern that yields better performance than random selection?

---

> ### Author Response · Authors · 2025-11-20
>
> > The proposed PENS reads as a combination of several existing ideas rather than a clearly novel contribution.
>
> We are happy to clarify our specific contributions.
> - **On novelty**: We adress this in detail in our general comment ["On the Novelty of PENS"](https://openreview.net/forum?id=EQuoft7Yc7&noteId=kvtUHKGxKz). In short, while others have added distance information, our novelty is being the first to use **PEs as the _sole_ spatial input**, demonstrating that this is sufficient to achieve SotA on large-scale instances, challenging the need for sparsification.
> - **On scaling strategy [4]**: The reviewer is correct that scaling strategies have been explored, but **[4] refers to ESF/SSMax, which is different from our contribution**. ESF modifies attention logits based on sequence length ($s \propto \text{log}(N)$) to help generalization. We already use this technique in our models. Our new contribution is a **coordinate rescaling** applied to the input city positions, to mitigate the embedding aliasing (and promote local bias with ALiBi). The two scaling methods are complementary.
>
> > Heavy dependence on tuning the scaling factor.
>
> We address the nature of the scaling factor in our general comment ["On the Scaling Factor and Large-Scale Generalization"](https://openreview.net/forum?id=EQuoft7Yc7&noteId=dpd6uJKudR). Crucially, **our method is not dependent on it**. As shown in Table 3 of the appendix, our PENS models are still highly competitive and generalize well even without any coordinate rescaling.
>
> > Insufficient experiments.
>
> We agree that demonstrating cross-task generalization is important and are currently conducting additional experiments on **CVRP**. We anticipate including these new results in the appendinx for the final version of the paper. We hope the reviewer will acknowledge our other points in the meantime.
>
> > PENS selects random origin and destination nodes. Is there a specific selection pattern that yields better performance than random selection?
>
> The choice of origin/destination indeed impacts the final solution quality. In our evaluations, we report the average optimality gaps over five independent runs. This ensures a fair and robust estimation.

---

### Official Review · Reviewer_pGYj · 2025-10-23

**Soundness:** 2
**Presentation:** 2
**Contribution:** 3
**Rating:** 4
**Confidence:** 5

**Summary:**

This paper explores the application and analysis of commonly used positional encoding techniques in NLP, such as ALiBi and RoPE, within the field of neural combinatorial optimization (NCO). The authors assert that by adapting these techniques to the Euclidean Traveling Salesman Problem (TSP), improvements are made in solving the problem. The proposed Positional Encoding-based Neural Solvers (PENS) achieve state-of-the-art results by leveraging these NLP techniques, particularly when scaling to large problem sizes such as TSP instances with up to 10,000 cities, without relying on graph sparsification.

**Strengths:**

1. The paper is innovative in applying NLP techniques to NCO, especially in combinatorial optimization problems like TSP. It’s a fresh perspective that could have broad applications.
2. The paper is generally well-written, and the experiments are presented in a structured manner. The figures, including the performance evaluations, are informative and contribute to understanding the results.

**Weaknesses:**

1. The focus on TSP is fine, but the approach should be tested on other combinatorial problems (like CVRP or FJSP) to see how generalizable it is.
2. The paper mainly adapts NLP techniques without making changes that would specifically suit NCO. This limits the paper's innovation.
3. The paper shows that increasing the scaling factor improves results, but selecting the right scaling factor still seems trial-and-error.
4. The methodology relies on an autoregressive approach, similar to NLP tasks, but the TSP's solution space has inherent differences from NLP, such as its cyclical nature. The direct transfer of sequence modeling techniques to this context may not always be suitable, and the authors should address how their approach adapts (or struggles) with combinatorial optimization problems that don't naturally fit the NLP paradigm.

**Questions:**

1. Are ALiBi and RoPE used in both the encoder and decoder?
2. In Table 1, PENS is much faster than BQ-NCO (similar architecture) on TSP-10,000. Can the authors explain why, particularly with PENS-R?

---

> ### Author Response · Authors · 2025-11-20
>
> We thank the reviewer for their detailed feedback and their recognition of our work's innovative application of NLP techniques to NCO.
>
> > The focus on TSP is fine, but the approach should be tested on other combinatorial problems (like CVRP or FJSP) to see how generalizable it is.
>
> We agree that demonstrating generalizability is important. We are currently conducting additional experiments on **CVRP** and anticipate including these results in the final version of the paper.
>
> > The paper mainly adapts NLP techniques without making changes that would specifically suit NCO. This limits the paper's innovation.
> >
> > The paper shows that increasing the scaling factor improves results, but selecting the right scaling factor still seems trial-and-error.
>
> This is a central point which we address in detail in our general comments. Our innovation lies in demonstrating how this adaptation is needed to solve a key NCO challenge. We invite the reviewer to read our general comments ["On the Novelty of PENS"](https://openreview.net/forum?id=EQuoft7Yc7&noteId=kvtUHKGxKz) and ["On the Scaling Factor and Large-Scale Generalization"](https://openreview.net/forum?id=EQuoft7Yc7&noteId=dpd6uJKudR) for the full discussion.
>
> > The direct transfer of sequence modeling techniques to this context may not always be suitable, and the authors should address how their approach adapts (or struggles) with combinatorial optimization problems that don't naturally fit the NLP paradigm.
>
> Our PEs operate within the attention layers and are **decoupled from the autoregressive scheme**. They could just as easily be used in non-AR models. We use the AR framework because it is a common and effective way to ensure constraint satisfaction in NCO. Our core contribution is independent of this choice.
>
> > Are ALiBi and RoPE used in both the encoder and decoder?
>
> Our architecture is a standard, decoder-only transformer. ALiBi and RoPE are applied within all layers of the single transformer stack.
>
> > In Table 1, PENS is much faster than BQ-NCO (similar architecture) on TSP-10,000. Can the authors explain why, particularly with PENS-R?
>
> This is a good observation. The reason is that our PENS implementation uses PyTorch's [`flex-attention`](https://pytorch.org/blog/flexattention/) to efficiently handle the custom attention mechanisms introduced by our work. We found this modern kernel to be well-optimized for RoPE's operations. BQ-NCO's implementation uses a less optimized kernel which accounts for the performance difference at the 10,000-city scale.

---

### Official Review · Reviewer_LAVY · 2025-10-30

**Soundness:** 2
**Presentation:** 3
**Contribution:** 2
**Rating:** 4
**Confidence:** 5

**Summary:**

The paper proposes a transformer-based neural solver for the Euclidean Traveling Salesman Problem (TSP) using positional encodings rather than coordinate projections. By adapting ALiBi and RoPE, two modern positional encoding methods originally designed for large language models, the authors introduce Positional Encoding-based Neural Solvers (PENS). The study demonstrates that these encodings provide inductive biases that improve the neural solver’s ability to generalize, particularly on large-scale TSP instances, achieving state-of-the-art results on problems with up to 10,000 cities. Furthermore, the paper proposes a coordinate rescaling technique to mitigate challenges arising from the increased density of city instances, further boosting performance.

**Strengths:**

1.The paper introduces a novel use of positional encodings (ALiBi and RoPE) in the context of the TSP, leveraging their ability to capture spatial relationships between cities. This innovation helps in scaling neural solvers to large TSP instances, outperforming previous methods that required graph sparsification.

2.The experimental results show that PENS achieves state-of-the-art performance, especially on large instances (up to 10,000 cities). It surpasses previous neural TSP solvers such as INViT and DGL, especially in terms of optimality gaps and computational efficiency.

3.The approach effectively handles large-scale instances of TSP without requiring sparsification, an improvement over prior works that rely on graph sparsification methods. The results on TSP-10,000 demonstrate that the method is capable of handling very large problem sizes efficiently.

**Weaknesses:**

1.Although the model achieves great results on large instances, the method still requires a full forward pass at each decoding step, which could be computationally expensive. Future work could focus on optimizing decoding efficiency and reducing computational overhead.

2.While the paper does a good job comparing positional encoding methods (ALiBi vs. RoPE), it could benefit from a deeper discussion on how other encoding techniques might be integrated with transformer-based solvers for TSP, such as sparse attention or graph-based encodings.

3.The paper uses a heuristic approach to estimate the best scaling factor for different TSP sizes, but it might be useful to provide a more rigorous method for determining the scaling factor, especially for different TSP variants, including those with asymmetric costs.

**Questions:**

1.How well does the method generalize to real-world TSP instances that may involve more complex constraints or asymmetric distances? Could PENS be adapted to handle such cases, and if so, how?
2.The rescaling factor significantly improves performance on larger instances. However, does this scaling factor have any impact on smaller TSP instances (under 1000 cities), and would it be better to use different scaling factors for different problem sizes?
3.Could the authors provide more detailed comparisons with non-transformer-based methods, especially those not relying on positional encodings or graph sparsification? How does the performance of PENS compare to other graph neural network-based approaches or reinforcement learning-based solvers for TSP?
4.While the model shows strong performance, the training time on large instances might be a limiting factor. Would incorporating techniques like gradient checkpointing, multi-GPU setups, or model pruning improve training efficiency without sacrificing accuracy?

---

> ### Author Response · Authors · 2025-11-20
>
> We thank the reviewer for their constructive feedback.
>
> > Future work could focus on optimizing decoding efficiency and reducing computational overhead.
>
> We agree. This is the trade-off of the autoregressive approach, which PENS inherits from BQ-NCO [1]. As the reviewer suggests, this is a promising line of future work, including techniques like quantization (e.g. GPTQ [2], speculative decoding [3]).
>
> > The paper uses a heuristic approach to estimate the best scaling factor for different TSP sizes, but it might be useful to provide a more rigorous method for determining the scaling factor, especially for different TSP variants, including those with asymmetric costs.
>
> We believe PENS-A is well-suited for the ATSP because it operates directly on the cost matrix. A simple adaptation would be to provide two views of the input: the original cost matrix $C$ and its transpose $C^T$. This would allow the model to learn relationships based on both "distance to" and "distance from" other cities. The rest of our method would remain unchanged and the scaling factor would be applied directly to the cost matrix.
>
> > does this scaling factor have any impact on smaller TSP instances (under 1000 cities)
>
> Yes, the optimal scaling factor is dependent on the problem size, but its impact diminishes for smaller instances. As we detail in the appendix (Table 3), the effect is marginal for TSP-500 and below.
>
> This supports our general comment ["On the Scaling Factor and Large-Scale Generalization"](https://openreview.net/forum?id=EQuoft7Yc7&noteId=dpd6uJKudR).
>
> > Could the authors provide more detailed comparisons with non-transformer-based methods, especially those not relying on positional encodings or graph sparsification? How does the performance of PENS compare to other graph neural network-based approaches or reinforcement learning-based solvers for TSP?
>
> We did include comparisons with SotA RL-based solvers. Both INViT and DGL are transformer-based, reinforcement learning-trained models.
>
> Comparisons to GNN-based methods like DIFUSCO [4] are challenging because they often rely on different decoding paradigms. For instance, DIFUSCO predicts edge probabilities and relies on computationally expensive decoding methods like MCTS to find a solution. This makes it difficult to isolate the architectural contribution. Our evaluation, which uses greedy decoding for all baselines, ensures a fair and direct comparison of the model architectures themselves.
>
> > Would incorporating techniques like gradient checkpointing, multi-GPU setups, or model pruning improve training efficiency without sacrificing accuracy?
>
> The reviewer is correct, these standard techniques would certainly help. We confirm our provided codebase already supports multi-GPU training.
>
> [1] BQ-NCO: Bisimulation quotienting for efficient neural combinatorial optimization, NeurIPS, 2023
>
> [2] GPTQ: ACCURATE POST-TRAINING QUANTIZATION FOR GENERATIVE PRE-TRAINED TRANSFORMERS, ICLR, 2023
>
> [3] A Theoretical Perspective for Speculative Decoding Algorithm, NeurIPS, 2024
>
> [4] DIFUSCO: Graph-based Diffusion Solvers for Combinatorial Optimization, NeurIPS, 2023

---

### Official Review · Reviewer_GQwc · 2025-10-30

**Soundness:** 2
**Presentation:** 3
**Contribution:** 2
**Rating:** 4
**Confidence:** 5

**Summary:**

This paper proposes Positional Encoding-based Neural Solvers (PENS), which introduce modern positional encodings (ALiBi, RoPE) into the neural combinatorial optimization (NCO) domain for solving the Euclidean TSP. Instead of projecting 2D coordinates into higher-dimensional spaces as in conventional NCO models, PENS leverages positional encodings as input representations. In addition, a coordinate rescaling scheme is incorporated, enabling the model trained solely on TSP-100 to generalize effectively to instances ranging from 100 up to 10K nodes.

**Strengths:**

1. The paper makes an innovative contribution by introducing an ALiBi-based positional encoding tailored for TSP, where Euclidean distances between nodes are used in place of token index distances.

2. The work explores and evaluates two forms of modern positional encodings (ALiBi and RoPE) in the context of TSP.

3. The overall presentation is clear, with a well-structured and accessible writing style.

**Weaknesses:**

1. **The experimental design is not sufficiently direct**. Since the key contribution is to introduce modern positional encodings (PE) for representing TSP inputs in transformer-based neural solvers, **it would be more convincing to explicitly replace the coordinate projection layers in classical NCO models** (e.g., AM-Kool [ICLR 2019], POMO) and recent strong baselines (e.g., BQ, LEHD) with the proposed positional encodings, while keeping the rest of the model unchanged. Such controlled experiments would clarify whether the improvements, particularly in generalization, stem from the PE themselves.

2. **The experimental results raise some concerns**. In Figure 4, the performance of **CoordNS** appears unusual. According to the paper (p.6, line 299), CoordNS essentially corresponds to a “**standard transformer backbone**.” Yet, when trained on TSP-100, it achieves only a **1.51%** gap relative to Concorde on TSP-1000—surpassing BQ, LEHD, and even [1], where [1] trains directly on TSP-1000 but still reports a **1.95%** gap. Such strong performance using raw coordinates alone seems noteworthy and warrants deeper investigation. It is unclear why the authors did not further analyze or discuss this unexpectedly strong result.

3. The paper **lacks sufficiently strong SOTA baselines**. The included comparisons with BQ, LEHD, and INViT are somewhat outdated. For instance, [1], which is also attention-based, achieves competitive or better performance: its **runtime** on TSP-1K is **far lower than PENS**, with only slightly worse performance, and on TSP-10K it outperforms PENS in **both runtime and solution quality**. While PENS is trained only on TSP-100 and generalized to 100–10K, whereas [1] is trained separately at each problem size, it would still be much more convincing to report results from stronger baselines under the same setting (trained on TSP-100 and tested across scales), including both **runtime** and **optimality gap**.

---

**Based on the above three points, I suggest that the authors conduct controlled experiments by replacing the linear projection layers in existing models with these two PEs, and then observe whether performance improves. This would provide stronger evidence for the effectiveness of PE itself.**

---

4. The experiments are **limited to TSP**, with no evaluation on other combinatorial optimization problems. It is unclear why the proposed approach cannot be applied to CVRP, for example. Is the limitation due to the fact that the two positional encodings used here cannot directly encode node demands as linear projections do? If the method is inherently restricted to TSP, **the broader significance of introducing positional encodings into NCO would be somewhat diminished**. I recommend that the authors at least **discuss how CVRP demands might be incorporated**. Furthermore, since ATSP only requires a distance matrix, additional **experiments on ATSP** (with [2], [3] as baselines) would strengthen the case for the general applicability and significance of PENS.

5. (minors) To improve rigor, the authors should avoid making absolute claims without sufficient literature coverage, even when qualified by phrases such as “to our knowledge.” For example, on p.2 line 99, the statement “the only ones to use distance matrices directly” is too strong. A simple literature search (or even directly using an LLM to deepresearch) may reveal additional NCO works that employ distance matrices [2,3,4]. Softening such claims would strengthen the paper’s credibility.


---

[1] Luo, Fu, et al. "Boosting neural combinatorial optimization for large-scale vehicle routing problems." The Thirteenth International Conference on Learning Representations. 2025.


[2] Kwon, Yeong-Dae, et al. "Matrix encoding networks for neural combinatorial optimization." Advances in Neural Information Processing Systems 34 (2021): 5138-5149.

[3] Pan, Wenzheng, et al. "UniCO: On unified combinatorial optimization via problem reduction to matrix-encoded general TSP." The Thirteenth International Conference on Learning Representations. 2025.

[4] Zhou, Changliang, et al. "ICAM: Rethinking Instance-Conditioned Adaptation in Neural Vehicle Routing Solver." (2025)

**Questions:**

1. The source of PENS’s performance, particularly its generalization ability, remains somewhat unclear. Could the authors provide additional experiments and discussion to clarify this point, especially in relation to Weaknesses 1 and 2?

2. How does the method perform on other combinatorial optimization problems? Please refer to Weakness 4 for details.

---

> ### Author Response · Authors · 2025-11-20
>
> We thank the reviewer for their detailed assessment of our work. We address the core concerns regarding experimental design, baseline strength, and the strong performance of our CoordNS baseline.
>
> > The source of PENS’s performance, particularly its generalization ability, remains somewhat unclear.
>
> The source of PENS's generalization is a combination of the PEs' inherent inductive biases and our scaling factor, addressing the two main issues in large-scale generalization identified by INViT. A detailed answer is provided in our two general comments ["On the Novelty of PENS"](https://openreview.net/forum?id=EQuoft7Yc7&noteId=kvtUHKGxKz) and ["On the Scaling Factor and Large-Scale Generalization"](https://openreview.net/forum?id=EQuoft7Yc7&noteId=dpd6uJKudR).
>
> > The experimental design is not sufficiently direct. [...], it would be more convincing to explicitly replace the coordinate projection layers in classical NCO models.
>
> While we did not replace PEs in _every_ classical model due to limited resources, our experimental setup is designed to isolate the contribution of PEs effectively: **Our baseline CoordNS serves as point of comparison**. Our CoordNS is already a highly competitive "classical NCO model", specifically derived from the SotA non-sparsifying architecture BQ-NCO [2], enhanced with modern transformer choices (prenorm/RMSNorm) and ESF/SSMax [3, 4].
>
> > The experimental results raise some concerns.
> >
> > the performance of CoordNS appears unusual
> >
> > [1] trains directly on TSP-1000 but still reports a 1.95% gap
>
> We agree that the competitive performance of CoordNS on TSP-1000 is noteworthy.
>
> As explained previously, our baseline is not a naive implementation but a strong configuration (BQ-NCO + ESF + architectural tweaks), which accounts for the strong performance. The good results of our baseline matches the one from BQ-NCO + ESF at that scale (1.51% gap vs 1.71%, Table 2), with the slight gain that we attribute to our contemporary transformer standards (prenorm/RMSNorm).
>
> We confirm that **CoordNS performance drastically decreases on TSP-10,000**, where PENS-A takes the definitive lead. Our analysis suggests that TSP-1000 represents the intermediate point where the feature-richness of the CoordNS projection is still viable, but lack of an explicit locality bias (which PENS-A provides).
>
> We believe the lower performance of [1] on TSP-1000 is due to its computationally lighter, sparse-attention mechanism, which likely limits the solver's capacity at a scale where our full-attention solvers still utilize the entire graph to produce better solutions.
>
> > The paper lacks sufficiently strong SOTA baselines.
>
> We only select methods trained on TSP-100 and tested across scales, for a fair comparison of the generalization ability. We stand by our current set of baselines (BQ-NCO, LEHD, INViT, DGL) as they represent the state-of-the-art across the categories of sparsifying and non-sparsifying methods. Notably, DGL has been published as recently as of last august, at IJCAI 2025.
>
> > The experiments are limited to TSP, with no evaluation on other combinatorial optimization problems.
>
> We agree that demonstrating cross-task generalizability is important.
>
> - **CVRP**: We are currently running additional experiments on CVRP. Our first results are positive and anticipate including these results in the final version.
> - **ATSP**: ATSP would be an excellent fit for PENS-A. Since ATSP only requires a distance matrix, the architecture remains largely unchanged. We suggest a slight adaptation by taking two parallel attention layers, one for the forward distance matrix, and one for the transposed. This would likely yield the best results as this lets cities know about their surroundings and which cities they are close to.
>
> > (minors) To improve rigor, the authors should avoid making absolute claims...
>
> We have followed the reviewer's advice and have corrected the related works section to cover the suggested articles, removing absolute claims.
>
> [1] Luo, Fu, et al. "Boosting neural combinatorial optimization for large-scale vehicle routing problems." The Thirteenth International Conference on Learning Representations. 2025.
>
> [2] BQ-NCO: Bisimulation quotienting for efficient neural combinatorial optimization, NeurIPS, 2023.
>
> [3] Scalable-softmax is superior for attention, 2025.
>
> [4] Improving generalization of neural vehicle routing problem solvers through the lens of model architecture. Neural Networks, 2025.

---

> > ### Comment · Reviewer_GQwc · 2025-11-24
> >
> > Thank you for the rebuttal. While it resolves part of my concerns, some remain unaddressed.
> >
> > ---
> >
> > 1. About the novelty, Weakness 1 and Question 1
> >
> > Regarding the claimed novelty, I find it challenging to conclude that the current experiments provide solid support for the three points you summarized.
> >
> > First, the results in Figure 4 do not clearly show that any specific PE consistently improves performance across all problem scales.
> >
> > Second, although your training setting differs from prior work, models in other papers still achieve better performance with substantially lower computational cost. For example, [1], which I mentioned earlier, also do not use sparsification. Therefore, the novelty statement you provided is overstated. At the very least, it should be qualified to something like "under the setting of training on TSP100 and generalizing to larger scales".
> >
> > Third, ESF originates from [5], so it is unclear why this should be counted as your contribution toward "bridging NLP and NCO".
> >
> > Taken together, the claimed novelty does not yet appear to be convincingly validated.
> >
> > Returning to the experimental setup: of course, it is not necessary to "replace PEs in every classical model". However, I maintain my original suggestion that you could select one or two classical models with relatively low training cost (such as AM or POMO, and the more models you include, the more convincing the results would be) and replace their input representations with NLP-style PE to test whether this yields performance gains. This would directly demonstrate whether your work contributes to "bridging NLP and NCO" at the input level, as compared to the standard approach of linearly projecting raw coordinates into high-dimensional embeddings.
> >
> > Without such experiments, your current method, which you described as "BQ-NCO [2] enhanced with modern transformer components (prenorm or RMSNorm) and ESF or SSMax [3,4]" plus PE, requires carefully designed ablation studies to determine where the improvements originate. For example, if most of the gains are due to ESF from prior NCO work and the impact of PE is minimal, then it would be difficult to regard the present work as making a meaningful contribution to NCO.
> >
> > ---
> >
> > 2. About other problems
> >
> > Thank you for adding the CVRP and ATSP experiments. I look forward to seeing their results.
> >
> > ---
> >
> >
> > [1] Boosting neural combinatorial optimization for large-scale vehicle routing problems. ICLR, 2025.
> >
> > [5] Improving generalization of neural vehicle routing problem solvers through the lens of model architecture. Neural Networks, 2025.

---

### Official Review · Reviewer_2M8Y · 2025-10-31

**Soundness:** 3
**Presentation:** 3
**Contribution:** 2
**Rating:** 4
**Confidence:** 3

**Summary:**

This paper presents PENS, a Transformer-based approach to solving the Euclidean TSP. Instead of feeding raw coordinates, it uses positional encodings — namely ALiBi and RoPE, which are commonly used in large language models — to represent spatial relationships between cities. The authors argue that this brings translation, rotation, and scale invariance, and helps models trained on small instances (TSP100) generalize to larger ones (TSP10,000). Results show that PENS performs well and even beats some sparsified Transformer baselines like INViT.

**Strengths:**

1. The paper is cleanly written and technically sound, with fair comparisons and detailed ablations.
2. It gives a practical insight: simple positional encodings can help Transformers scale better for geometric problems.
3. The results are solid and the method is easy to reproduce.

**Weaknesses:**

1. The main issue is limited novelty. The paper basically transfers ALiBi and RoPE (well-known in NLP) to TSP. There’s no new learning idea or inductive bias proposed.
2. The improvement seems mostly empirical, driven by better coordinate scaling and heuristics rather than a truly new model concept.
3. It doesn’t help us understand neural combinatorial optimization better. There’s no new training strategy or learning dynamic introduced.
4. Evaluation is limited to synthetic Euclidean TSP. It’s unclear how it performs on other routing problem types.

**Questions:**

1. Could you provide a clearer theoretical or analytical explanation of why ALiBi or RoPE leads to better translation, rotation, or scale invariance in TSP?
2. The experiments are only conducted on synthetic Euclidean TSP datasets. Have you tested the model on non-Euclidean graphs or other routing problems (such as VRP)?

---

> ### Author Response · Authors · 2025-11-20
>
> We thank the reviewer for their careful reading and comments.
>
> > The main issue is limited novelty. The paper basically transfers ALiBi and RoPE (well-known in NLP) to TSP. There’s no new learning idea or inductive bias proposed.
> >
> > The improvement seems mostly empirical, driven by better coordinate scaling and heuristics rather than a truly new model concept.
> >
> > It doesn’t help us understand neural combinatorial optimization better. There’s no new training strategy or learning dynamic introduced.
>
> We respectfully disagree that our contribution is purely empirical or lacks insight into NCO. Our work's primary contribution is architectural and provides a powerful and explicit inductive biases (locality, invariances) and draws a direct link between the NLP and NCO domains. See a more detailed answer at our general comment ["On the Novelty of PENS"](https://openreview.net/forum?id=EQuoft7Yc7&noteId=kvtUHKGxKz).
>
> > Evaluation is limited to synthetic Euclidean TSP. It’s unclear how it performs on other routing problem types.
> >
> > The experiments are only conducted on synthetic Euclidean TSP datasets. Have you tested the model on non-Euclidean graphs or other routing problems (such as VRP)?
>
> We acknowledge the need for broader evaluation. We already include results on **TSPLIB** (real-world, non-synthetic instances). Furthermore, to demonstrate the generalizability of our method, **we are currently conducting experiments on CVRP**, for which early results are competitive with other SotA approaches. We anticipate including these new results in the final version of the paper.
>
> > Could you provide a clearer theoretical or analytical explanation of why ALiBi or RoPE leads to better translation, rotation, or scale invariance in TSP?
>
> We are happy to clarify the theoretical basis for PEs' geometric properties:
> - **ALiBi**: The core input to ALiBi is derived from the distance matrix. The distance matrix is inherently invariant to translation, rotation, and reflections of the city coordinates. This property is thus inherited directly by the neural solver.
> - **RoPE**: RoPE encodes relative positioning. The dot-product between a key and query is proportional to the difference between their positional vectors, making the attention result dependent only on the _relative difference_ between cities, thus achieving translation-invariance.
>
> We do not make any claim about scale invariance in the paper. The necessity of our scaling heuristic (discussed in our general comment ["On the Scaling Factor and Large-Scale Generalization"](https://openreview.net/forum?id=EQuoft7Yc7&noteId=dpd6uJKudR)) precisely demonstrates this remaining challenge in current neural solvers.

---

### Author Response · Authors · 2025-11-20
**On the Novelty of PENS**

We thank the reviewers for their thoughtful feedback on this point. We wish to clarify our novel contribution, which is not the invention of the components (ALiBi, RoPE) but their **novel and highly effective application to solve a key challenge in neural combinatorial optimization**.

Our novelty is threefold:
- **A new input paradigm**: Our work is the *first to demonstrate* that modern positional encodings (PEs) can serve as the *_sole_ spatial input* for a neural TSP solver. While prior works have added distance-based biases [1], they still rely on projecting raw coordinates and crafting additional features. We show that _replacing_ coordinate projections entirely with PEs is not only viable but superior. This is a non-trivial result.
- **SotA on large-scale without sparsification**: Our most significant finding is that this simple approach **achieves state-of-the-art results on large-scale instances**, an area previously dominated by methods requiring complex hand-crafted graph sparsification. On TSP-10,000, we reach 5.46% against the previous 7.05% previous SotA result. This directly challenges the prevailing assumption that sparsification is necessary for large-scale generalization.
- **Bridging NCO and NLP**: Our paper deliberately anchors this NCO solution in the established NLP literature. This connection is significant because it shows that key NCO challenges (like problem-size generalization) have direct parallels in NLP (length extrapolation).
  - For example, we adopted Scalable-softmax (SSMax) [2] from NLP, which we noted was concurrently developed as ESF [3] in the NCO literature.
  - This demonstrates that NCO researchers can directly benefit from advances in the NLP domain rather than re-discovering parallel solutions.

We believe our idea of adapting NLP PEs to spatial positions encountered in NCO is original and provides insightful results about the field in general. Our simple approach challenges the complex SotA methods.


[1] Distance-aware attention reshaping for enhancing generalization of neural solvers. TNNLS, 2025.

[2] Scalable-softmax is superior for attention, 2025.

[3] Improving generalization of neural vehicle routing problem solvers through the lens of model architecture. Neural Networks, 2025.

---

### Author Response · Authors · 2025-11-20
**On the Scaling Factor and Large-Scale Generalization**

A truly scale-invariant solver that requires no adjustment remains the ultimate goal for NCO, and we view this as important future work. However, rescaling the coordinates is more than a heuristic, it is a finding that **mitigates a fundamental bottleneck** in current neural solvers.

The INViT paper [1] identified two key issues in large-scale generalization:
- _Interference from irrelevant nodes._
- _Embedding aliasing._

The scaling factor is both a solution and an experimental measure:
- **Targeting aliasing**: By spreading the city coordinates, we effectively lower the density and make it easier for the model to distinguish between close neighbors.
- **Significance**: This simple transformation **divides by two the optimal gap** on TSP-10,000, offering a strong estimation of how much performance is currently hidden by these density-related mismatches.
- **Synergy with PENS-A**: This technique uniquely complements PENS-A (ALiBi) because the increased relative distances further amplify ALiBi's soft locality bias, helping the solver focus on the relevant neighborhood. This phenomenon is illustrated in Figure 6 of the appendix, where PENS-A is the only model that remains consistent under multiple scaling factor values.

Moreover, the fact that the **exact same model weights** can successfully solve TSP-10,000 instances simply by rescaling the input coordinates demonstrates that:
- **The reasoning capacity exists**: The transformer has the computational power and algorithmic logic required to solve these instances without requiring hardcoded sparsification.
- **The density bottleneck**: We experimentally show that embedding aliasing is of great importance regarding large-scale generalization. This result should motivate focused research toward finding new and better approaches.

We initially hypothesized that the scaling factor should maintain constant spatial density relative to the training set, following the rule $s \propto \sqrt{n_{test}/n_{train}}$. However, our experiments revealed a non-trivial relationship as the empiricaly optimal factors deviate from this theoretical baseline.
One possibility that we envisage is to investigate the problem from the point of view of **discrepancy theory**. While the density of the problem generation is uniform, the actual realization of points inevitably contains local irregularities. Deriving a principled scaling formula that accounts for these probabilistic phenomena may be a promising avenue for future work.

We emphasize that **PENS's competitive performance is not solely reliant on this heuristic**. Even without the scaling factor, our results remain strong, confirming the architectural benefits of using PEs. A detailed comparison can be found in Table 3 of the appendix.

[1] INViT: A generalizable routing problem solver with invariant nested view transformer, ICML, 2024.

---

### Comment · Area_Chair_rwtf · 2025-11-25
**Violation of the anonymity rules**

Dear Authors,

I am sorry to inform you, you have uploaded a revised paper with explicit author information, which violates the anonymity rules. We have to reject this submission.

Regards,
AC

---

### Author Response · Authors · 2025-12-03
**Summary for the Area Chair (AC)**

Dear AC,

We appreciate your time and effort in dealing with our submission, particularly
given the unprecedented circumstances. To facilitate your decision, we provide
a concise summary of the technical review process and a formal request for
reconsideration of the initial administrative rejection.

## Summary of the Reviews

The initial reviews were unanimously constructive, with **all reviewers providing
a rating of 4**. All reviewers acknowledged the paper's strong empirical results
and clear presentation but raised core concerns regarding the novelty of the
approach and the reliance on the coordinate rescaling heuristic.

**Key Contributions and Responses to Novelty Concerns**

We addressed the concerns about novelty and the foundational input paradigm in
our general comment ["On the Novelty of PENS"](https://openreview.net/forum?id=EQuoft7Yc7&noteId=kvtUHKGxKz).

- **Novel Input Paradigm**: Reviewers noted that our method adapts known
  positional encodings (**ALiBi** and **RoPE**) from the NLP domain to NCO. Our
  contribution lies in demonstrating that these PEs can successfully serve as
  the **sole spatial input** for a neural TSP solver, resulting in a highly
  competitive model.
- **Inductive Bias:** The choice of ALiBi and RoPE is motivated by known
  challenges in NCO concerning **scaling to large instances**. We showed that
  models using our Positional Encoding-based Neural Solvers (PENS) compare
  favorably against baselines that rely on city coordinates, confirming the
  architectural benefits of PEs' inherent locality and invariance biases.
- **Bridging NCO and NLP:** We frame the challenge of generalizing to large-scale
  instances in NCO as the **length extrapolation problem** in NLP, illustrating
  how concepts from one domain can provide powerful insights and solutions for
  the other.

**Justification of the Rescaling Heuristic**

The importance of the coordinate scaling factor was discussed in detail in our
general comment ["On the Scaling Factor and Large-Scale Generalization"](https://openreview.net/forum?id=EQuoft7Yc7&noteId=dpd6uJKudR).

- **Mitigating Bottlenecks:** We introduced the simple rescaling heuristic to
  explore the impact of **embedding aliasing**, as known density bottleneck in
  large-scale NCO generalization.
- **Empirical Impact**: The heuristic proved highly effective, revealing the
  critical importance of tackling density-related issues for large instances.
  When couped with our PENS architecture, we achieved **SotA results on TSP without requiring graph
  sparsification**, a technique previously considered necessary for generalization at this scale.


## Reinforcing the Scope with Additional Experiments

The reviewers unanimously requested broader validation beyond the TSP.

- **Generalizability:** To demonstrate the generalizability of our method to
  other routing problems, the reviewers suggested testing on the **Capacitated Vehicle Routing
  Problem**.
- **Commitment:** We initiated these experiments during the rebuttal phase and
  are committed to including the new results in the final version of the paper.


In conclusion, we believe that we comprehensively addressed the technical
questions raised by the reviewers regarding novelty, methodology and
generalizability, and were on track to refine our manuscript with the key
additional results requested. We hope this summary of the technical discussion
supports a re-evaluation of the submission.

---

### Meta-Review · Area_Chair_yX53 · 2026-01-02

**Summary:**

This paper was desk-rejected by the initial AC due to a violation of anonymity rules (upload of a revised manuscript containing author information) prior to the OpenReview incident. I fully agree with the initial AC's decision, regardless of the subsequent OpenReview incident. However, as requested by the authors and in light of the current situation, I provide the following summary of the review process along with my recommendation for this submission.

This work proposes PENS, a positional encoding-based neural solver, for neural combinatorial optimization (NCO). The key contribution of PENS is to introduce advanced positional encoding methods (ALiBi and RoPE) originally developed in the NLP community to NCO. A heuristic rescaling approach is also proposed to further improve the neural solver's performance, especially on large-scale problem instances. Experimental results show that PENS, only trained on TSP instances with 100 nodes, can achieve promising performance on TSP instances with up to 10K nodes.

All reviewers initially assigned negative scores (4, 4, 4, 4, 4) to this work and raised many concerns regarding its novelty, the contribution of advanced positional encoding to NCO, the heuristic and trial-and-error nature of the scaling factor, certain claims and statements, experimental design and setup, ablation studies, and experimental evaluation on other combinatorial optimization problems. The authors provided a rebuttal to address these concerns. However, after the rebuttal, many of the major concerns remain. Reviewer GQwc believes some important claimed contributions are still not well supported, and the other reviewers did not respond to the rebuttal. Finally, all reviewers maintained their initial negative scores.

I have read this paper in detail. While I also find the work well-written and easy to follow, and the proposed PENS method can achieve promising results, I share the reviewers' concerns regarding the contribution, novelty, experimental setup, and evaluations of the current manuscript. I believe the reviewers would have maintained their initial scores (4, 4, 4, 4, 4)  even if they had been able to participate fully in the discussion.

Therefore, I recommend rejecting this work.

**Reviewer Concerns:**

While I believe the rebuttal may partially address some issues, the major concerns still remain:

**Contribution:** I agree with Reviewer GQwc that the true contribution and impact of positional encoding (PE) is not well supported by the current experiments and analysis. The PENS-A/R/AR models incorporate multiple advanced components compared to prior NCO models (e.g., prenorm/RMSNorm, ESF, SSMAX, flex-attention, and PE), which make the isolated effect of PE insufficiently ablated.

The rebuttal claims that "the good results of our baseline [CoordNS without PE] matches the one from BQ-NCO + ESF at that scale (1.51% gap vs 1.71%, Table 2)". However, BQ-NCO + ESF can achieve a 15.12% gap on TSP-10K, which outperforms PENS-R without the scaling factor (16.73%), whereas CoordNS has a larger gap at 23.47%. These results indicate that the CoordNS baseline is not equivalent to BQ-NCO + ESF.

To clearly demonstrate the contribution of PE to NCO, I would recommend adding PE directly to the original BQ-NCO model without other advanced modules and evaluating the direct performance improvement. Experiments that incorporate PE into other NCO models (e.g., AM, POMO, LEHD), as suggested by Reviewer GQwc, could also meaningfully assess PE's generalization ability.

**Novelty:** Given that PE itself is not novel, a clear and detailed analysis of how it improves NCO performance is crucial to support the contribution of this work, as emphasized by multiple reviewers. Although attention visualizations for different scaling factors are provided in the Appendix, a more thorough theoretical or empirical analysis of PE is needed, especially since no single PE method consistently achieves the best performance across different problem sizes.

**Generalization to Other Problems:** All reviewers requested comparisons on combinatorial optimization problems beyond TSP. The authors mentioned having initiated such experiments, but did not provide any results in the rebuttal. A comprehensive and fair evaluation on other problems (e.g., CVRP, ATSP) would significantly strengthen the contribution of this work.

**Accurate Claims and Statements:** Certain claims and statements still require careful revision. Please refer to the detailed comments from Reviewer GQwc.

**Reviewer Scores:**

Based on the discussion above, I believe the reviewers would have maintained their initial scores even if they had been able to participate fully in the discussion.

---

### Decision · Program_Chairs · 2026-01-26

Reject